**PLOS** NEGLECTED TROPICAL DISEASES

# Real life condition evaluation of Inoserp PAN-AFRICA antivenom effectiveness in Cameroon

**Jean-Philippe Chippaux**[1,2]*, **Rodrigue Ntone**[3], **David Benhammou**[4], **Yoann Madec**[4], **Gaëlle Noël**[2], **Anais Perilhou**[2], **Fai Karl**[3], **Pierre Amta**[5], **Marie Sanchez**[6], **Lucrece Matchim**[3], **Pedro Clauteaux**[2], **Lucrèce Eteki**[3], **Mark Ndifon**[3], **Yap Boum**[3], **Armand S. Nkwescheu**[7], **Fabien Taieb**[8]

**1** Université Paris Cité, Institut de Recherche pour le développement, MERIT, Paris, France, **2** Institut Pasteur, Université Paris Cité, Centre de Recherche Translationnelle, Paris, France, **3** Epicentre Yaoundé, Yaoundé, Cameroon, **4** Institut Pasteur, Université Paris Cité, Emerging Diseases Epidemiology unit, Paris, France, **5** Tokombéré Hospital, Tokombéré, Cameroon, **6** Institut Pasteur, Université Paris Cité, Data management core facility, Paris, France, **7** Cameroon Society of Epidemiology, Yaounde, Cameroon, **8** Institut Pasteur, Medical Center, Paris, France

* jean-philippe.chippaux@ird.fr

**Data Availability Statement:** Data available at DOI: 10.5281/zenodo.8200198 (md5:2923b05c239f493b4dd1d4f54163f574).

## Abstract

### Background

Snakebites is a serious public health issue but remains a neglected tropical disease. Data on antivenom effectiveness are urgently needed in Africa. We assessed effectiveness of Inoserp PAN-AFRICA (IPA), the recommended antivenom available in Cameroon.

### Methodology/Principal findings

We enrolled 447 patients presenting with snakebite in 14 health facilities across Cameroon. At presentation, cytotoxicity, coagulation troubles and neurotoxicity were graded. We administered two to four vials of antivenom to patients based on hemotoxic or neurotoxic signs. We renewed antivenom administration to patients with persistence of bleedings or neurotoxicity 2 hours after each injection. We defined early improvement as a reduction of the grade of envenomation symptoms 2 hours after first injection. Medium-term effectiveness was investigated looking at disappearance of symptoms during hospitalization. After hospital discharge, a home visit was planned to assess long-term outcomes.

Between October 2019 and May 2021, we enrolled 447 (93.7%), including 72% from the savannah regions. The median [IQR] age was 25 [14–40]. Envenomation was diagnosed in 369 (82.6%) participants. The antivenom was administered to 356 patients (96.5%) of whom 256 (71.9%) received one administration. Among these patients, cytotoxic symptoms were observed in 336 (94.4%) participants, coagulation disorders in 234 (65.7%) participants and neurotoxicity in 23 (6.5%) participants. Two hours after the first administration of antivenom, we observed a decrease in coagulation disorders or neurotoxicity in 75.2% and 39.1% of patients, respectively. Complete cessation of bleedings and neurotoxicity occurred in 96% and 93% of patients within 24 hours, respectively. Sequelae have been observed in 9 (3%) patients at the home visit 15 days after hospital admission and 11 (3%) died including one before antivenom injection.

**Funding:** The study was funded by the Institut Pasteur. Inosan Biopharma contributed to the financing through a grant paid to the Institut Pasteur. Inosan Biopharma had no role in study design, data collection and analysis, decision to publish, or preparation of the manuscript.

**Competing interests:** The authors have declared that no competing interests exist.

## Conclusions/Significance

We confirmed good effectiveness of the IPA and highlighted the rapid improvement in bleeding or neurotoxicity after the first administration. Sequential administrations of low doses of antivenom, rigorously assessed at short intervals for an eventual renewal, can preserve patient safety and save antivenom.

## Trial registration

NCT03326492.

## Author summary

Snakebite envenomation is a public health issue in all sub-Saharan countries. Their management remains a challenge due to the high cost of antivenom and complex treatment-seeking behavior.

The objective of this study was to evaluate the tolerance and effectiveness of a commonly used antivenom in Cameroon, in 14 sites representative of the diversity of common epidemiological situations in sub-Saharan Africa. The treatment protocol was that recommended by the Cameroonian Ministry of Health. We reported in the present manuscript results on antivenom effectiveness.

Administration of IPA (at least two vials) was decided in all patients presenting with any symptoms of envenomation (cytotoxicity, bleeding, neurotoxicity) regardless of severity. Two to four vials of antivenom were administered to patients depending on whether they had coagulopathy or neurotoxic disorders, respectively. We repeated the administration of antivenom at the same dose to patients if hemorrhagic or neurotoxic signs persisted 2 hours after each injection.

During 20 months, we examined 477 patients and enrolled 447 (94%). Three hundred fifty-six patients presenting envenomation signs have received at least one dose of antivenom. Envenomation was diagnosed in 369 (83%) participants, out of which, 9 (3%) kept sequelae of varying severity, and 11 (3%) died, including one before the antivenom injection. Cytotoxic symptoms were observed in 336 (94.4%) participants, coagulation disorders in 234 (65.7%) participants and neurotoxic syndrome in 23 (6.5%) participants. A single antivenom administration was performed for 256 (71.9%) patients. Two hours after the first administration of antivenom, coagulation disorders and neurotoxicity decreased in 75.2% and 39.1% of patients, respectively. Complete stop bleedings and neurotoxicity occurred in 96% and 93% of patients within 24 hours, respectively.

We confirmed the good effectiveness of IPA and highlighted the rapid improvement in bleedings or neurotoxicity after its first administration.

## Introduction

Snakebite envenomation (SBE) is an actual public health issue in most tropical countries, recently added to the list of neglected tropical diseases by the World Health Organization (WHO) [1,2]. In sub-Saharan Africa (SSA), more than one million snakebites occur yearly, resulting in at least 500,000 SBEs and 30,000 deaths. SBE affects poor rural populations, mainly during agricultural and pastoral works in areas where snakes are attracted by favorable environmental conditions, including places for their security, mating, breeding and preys [3–6].

Beyond the high mortality, snakebites have a considerable socio-economic impact linked to the consequences of SBE, which can lead to functional disability or amputation [7–9].

In SSA, two main families of venomous snakes are responsible for most SBEs: Viperidae and Elapidae. Viperidae (mainly *Echis romani*–formerly *E. ocellatus*–and *Bitis arietans*, two deadly species present in northern Cameroon, and *Bitis gabonica*, *B. nasicornis* or *Atheris* spp. present in central and southern Cameroon) possess enzyme-rich venoms that causes inflammation, coagulation disorders and necrosis. The venoms of Elapidae (cobras present in all Cameroon and green mamba in South Cameroon) mainly include toxins, resulting in neurotoxicity that leads to paralysis of muscles, particularly respiratory muscles, and phospholipases $A_2$ occasioning necrosis.

Antivenom is the only etiologic treatment of SBE. The main problem encountered in SSA is the poor accessibility and/or availability of antivenoms [5]. The causes are multifactorial but dominated by the cost of the product, the poverty of most patients, the absence of public or private funding, poor knowledge of its use by health personnel, the preference of victims for traditional medicine [3,5,10–12]. For many years, the standard treatment for SBE in SSA was IPSER-Africa, then FAV-Africa, manufactured by Sanofi Pasteur. In 2014, the manufacturer stopped the production because it was not profitable [11]. The need for antivenom in this region has led to the development of several antivenoms [13], including Inoserp PAN-AFRI-CAN (IPA) manufactured by Inosan Biopharma, which covers at least 18 snake species present in SSA. This antivenom is currently used in SSA and few clinical studies showed its usefulness [13–15].

The efficacy of an antivenom should be evaluated using a randomized clinical trial comparing the results of a group of patients treated with the experimental antivenom versus either a placebo or a reference antivenom whose efficacy is formally established [16]. The placebo is not currently approved by health authorities and ethics committees due to the risks faced by the untreated group. Furthermore, there is no consensus reference antivenom for SSA. The effectiveness of an antivenom can also be compared to a historical control group [17,18]. However, the limitations of historical control groups are well known. Case series are most often retrospective with non-standardized criteria and outcomes, which reduces their reliability. In addition, the results are imprecise and subject to significant bias.

Irrespective to the neutralizing capacity, the effectiveness of the antivenom depends on many extrinsic factors, including the quantity and toxicity of the venom, delay and quality of patient care, and proper use of the antivenom. These factors are complex or even impossible to control with the techniques available in the peripheral health facilities of SSA, the very ones which receive up to 95% of the SBE.

In addition, the effective dose is difficult to anticipate. It is generally based on a) the neutralizing capacity of the antivenom, estimated from preclinical tests in mice and therefore difficult to transpose to humans [19] and, b) the average quantity of venom that each species is capable of inoculate, which is largely speculative–not to mention intraspecific variations in venom [20]. The initial dose recommended, in particular by the WHO and most antivenom manufacturers, varies between 4 and 20 vials depending on the characteristics of the antivenom and the species responsible for the bite, dose to be renewed if necessary [21]. Applied to all SBEs, this strategy is costly and very consuming of the scarce antivenom resource. Anyway, the recommendation is usually not enforced simply because most patients are reluctant to purchase such amounts of antivenom, especially if their perceived health condition is mild.

Consequently, the sequential immunotherapy strategy makes it possible to envisage substantial savings while respecting usual practices. It consists of administering a low initial dose (2 vials for Viperidae bites and 4 vials for Elapidae bites). In most cases, this dose is sufficient. However, some patients require higher doses. It is therefore advisable to quickly evaluate the

effect of the first dose to renew it if necessary. Several evaluations of antivenom have been conducted under similar conditions but none on this scale, nor with this specific objective [14,15,18,22].

The study "evaluation of antivenom in Africa" (Evaluation du Sérum Antivenimeux en Afrique, ESAA) aimed to assess tolerance and effectiveness of IPA in real-life conditions in Cameroon, and identify the factors associated with the effectiveness of the antivenom. In the present manuscript, we presented results on the effectiveness outcomes collected during the study. Results presenting outcomes based on tolerance has been analyzed separately will be presented in a near future dedicated article.

## Methods

### Ethics statement

The ESAA study was registered on clinicaltrial.gov (NCT03326492). Ethical clearance was obtained from the Institut Pasteur Institutional Review Board (Paris, N° IRB00006966/2017-06) and the National Ethics Committee of Cameroon (Yaoundé, N°2018/03/994/CE/CNERSH/SP). The study was authorized by the Ministry of Public Health of Cameroon (N° 631–18.18).

Formal written consent was obtained from all patients participating in the study, including children for whom consent was obtained from the parent/guardian who brought the child to the hospital. However, patients admitted in the participating study sites and who did not consent to participate to the study received care according to the national protocol, i.e. an antivenom available at the Health center level and local treatment. Indications for administration and re-administration were carried out according to national recommendations and were based on exchanges between the health care staff and the bitten patient. No data were collected for patients who did not consent to participate in the study.

A scientific committee held two meetings to examine the results, in particular severe adverse events. This committee was composed of the main investigators from Paris and Yaoundé, a representative of the sponsor (Institut Pasteur, Paris) and two independent international experts.

### Population

The ESAA project is an observational prospective clinical study implemented in 14 health centers in Cameroon. These health centers were distributed throughout the country, from the Far-North/North/Adamawa (North Cameroon (NC)) to the South-West/South-Centre (South Cameroon (SC)) (Fig 1). They were selected by the Cameroonian health authorities according to the mean incidence of snakebites and ecological environment in order to cover a wide range of snake ecology, which differs between the savannah area in the NC, the forest area and forested low mountain in SC.

Participants were enrolled among patients who consulted the participating sites for snakebite, with or without envenomation, between October 25, 2019, and May 3, 2021. The inclusion criteria were, a) having been bitten by a snake (with or without clinical symptoms), b) being five years of age or older, c) not having a known allergy to antivenom, d) not having received antivenom between the bite and presentation to the hospital, and e) signing the informed consent form.

Antivenom and symptomatic treatment were provided free of charge to the patients included in the study.

Data were first collected on a paper CRF and then entered into the REDCap data collection software [23,24]. Socio-demographic data, snakebite circumstances, time and clinical

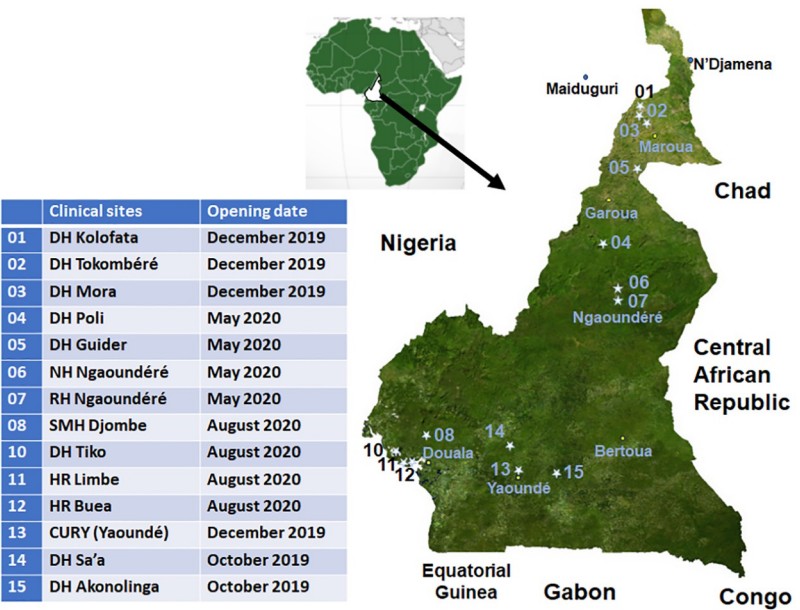

| | Clinical sites | Opening date |
|---|---|---|
| 01 | DH Kolofata | December 2019 |
| 02 | DH Tokombéré | December 2019 |
| 03 | DH Mora | December 2019 |
| 04 | DH Poli | May 2020 |
| 05 | DH Guider | May 2020 |
| 06 | NH Ngaoundéré | May 2020 |
| 07 | RH Ngaoundéré | May 2020 |
| 08 | SMH Djombe | August 2020 |
| 10 | DH Tiko | August 2020 |
| 11 | HR Limbe | August 2020 |
| 12 | HR Buea | August 2020 |
| 13 | CURY (Yaoundé) | December 2019 |
| 14 | DH Sa'a | October 2019 |
| 15 | DH Akonolinga | October 2019 |

**Fig 1. Location of centers participating in the ESAA study [The map was obtained from Wikipedia, https://fr. wikipedia.org/wiki/Cameroun#/media/Fichier:Cameroon_sat.png, CC0 1.0].**

presentation at the time of hospital arrival were collected. In case of antivenom injection, clinical data were collected at the time of administration, and during hospitalization until discharge as described below.

## Sample size

There was no pre-established sample size for the effectiveness analysis, as the main objective of the study was to document the tolerance of the antivenom.

## Snake identification

The snake responsible for the bite was formally identified when the snake was brought to the hospital by the patient or family, or when a photo had been taken. Photos of the snake were sent to an expert (JP Chippaux) to identify the species using published keys and descriptions [25,26]. In the absence of the snake, the patient was asked to examine photos of the main snakes in Cameroon arranged on a large A3 poster to allow for identification that was confirmed by specific symptomatology, e.g., hemorrhage due to *Echis romani* envenomation in NC.

## Envenomation and clinical follow-up

SBE grade assessment is detailed on S1 Appendix. SBE was defined by the presence of at least one of the following symptoms at admission to hospital: a) edema with a grade equal to or greater than 1, b) bleeding with a grade equal to or greater than 1, and/or a whole-blood clotting time on dry tube at 20 minutes (WBCT20) with a grade equal to or greater than 1 reflecting a coagulation abnormality to varying degrees (incomplete, friable or absent clot; S2 Appendix), and/or c) signs of neurotoxicity with a grade equal to or greater than 2. If the patient presented none of these signs at hospital admission, another evaluation was carried out

two hours later. SBE was considered if at least one of the symptoms described above was observed at the second evaluation, otherwise SBE was ruled out.

Patients were clinically assessed 2 hours after the initial injection, and if new injections were needed, they were assessed again 2 hours after that injection. Evaluations were also scheduled 12, 24, 48 hours, 3 days after the last injection and at the moment of hospital discharge. After hospital discharge, a home visit was scheduled at day 15 to assess long-term outcomes.

## Antivenom administration

IPA manufactured by Inosan Biopharma is the reference antivenom currently available in Cameroon. It is a lyophilized polyvalent antivenom composed of highly purified immunoglobulin fragments (F(ab')$_2$) produced by immunizing horses with the venoms of 14 species of snakes *Echis ocellatus*, *E. pyramidum*, *E. leucogaster*, *Bitis gabonica*, *B. nasicornis*, *B. arietans*, *Naja haje*, *N. melanoleuca*, *N. nigricollis*, *N. pallida*, *Dendroaspis polylepis*, *D. viridis*, *D. angusticeps* and *D. jamesoni*). This antivenom is effective against eighteen species of snake due to its para-specificity for species whose venom is not included in the horse immunization protocol (S3 Appendix). Each vial contains no more than 1 g of total proteins and neutralize at least 250 LD$_{50}$ of the venom of *E. ocellatus*, *B. arietans*, *N. nigricollis* and *D. polylepis*. IPA from a single batch (#8IT11001; expiration date Nov. 2021) was used for all patients at all study centers.

Storage conditions of vials, in particular room temperature, was controlled and was not to exceed 30°C (86°F), exceptions were permitted up to 40°C (104°F) for a maximum period of 6 months. A storage temperature monitoring system was put in place in all 14 study sites.

Administration of IPA was performed through intravenous injection either by direct intravenous injection of the 10 ml solution slowly over a minimum of 3 minutes per vial, or through a drip (2 vials of 10 ml reconstituted solution diluted with 50 ml of sterile isotonic saline solution) over 30 minutes. Patient care followed local standard of care and recommendations of both the manufacturer and algorithm recommended by Cameroonian ministry of Public Health (S3 Appendix). However, we enhanced application of previously exposed recommendation especially in terms of doses administration and intervals between clinical examinations.

Patients with SBE who presented edema or coagulation disorders (bleedings and/or abnormal 20WBCT) received two vials, while those who presented neurotoxicity received four vials. Two hours after the initial antivenom injection, a clinical examination was carried out; if bleedings or neurotoxicity persisted, worsened or appeared, a new antivenom administration following the same specifications as the first one (i.e 2 or 4 vials in case of persistence of bleedings or neurotoxicity, respectively) was performed and a clinical evaluation was conducted two hours later. This procedure applied until the signs disappeared (S3 Appendix).

## Statistical analysis

Effectiveness was investigated in all patients with envenomation symptoms who received at least one antivenom injection and who had at least one clinical evaluation following the injection. Early effectiveness was defined as a grade reduction of the envenomation sign 1 to 2 hours after the first injection, confirmed at a second evaluation conducted 3 to 4 hours after the first injection. Early effectiveness was investigating in patients with coagulation disorders or neurotoxicity. Due to the well-known slow decrease of edema, early effectiveness was not regarded for cytotoxic signs. Factors associated with early effectiveness were identified using logistic regression models.

Medium-term effectiveness was investigated looking at disappearance of envenomation sign during hospitalization (i.e. from H2 until hospital discharge). This analysis was conducted in patients with edema, in patients with bleedings (those who presented only abnormal

20WBCT were not considered as 20WBCT was not performed routinely beyond 2 hours after the initial injection), and in patients with neurotoxicity. Time to disappearance of envenomation sign was described using Kaplan-Meier estimates. Factors associated with medium-term effectiveness were identified using an accelerated failure time model with log-normal distribution as the proportional hazard assumption was not valid for some variables.

The following factors were considered in all univariate analyses: gender, age, *Body Mass Index* (BMI) for those over 20 years old, time between snakebite and antivenom administration, use of one or more traditional medicine (scarification, black stones, tourniquet, induced vomiting, medicinal herbs, traditional bandage), regions where snakebite occurred, Glasgow score, edema, bleeding and neurotoxicity gradings. For all statistical analysis described above, factors associated with the outcome with a p-value < 0.20 in univariate analysis were considered in the multivariate analysis. Then, a backward stepwise procedure was applied. All analyses were performed using Stata 17 (Stata Corp., College Station, TX, USA).

## Results

### Demographics

During the 18-month study enrolment period, 477 patients presented to one of the 14 study sites for snakebite, out of whom 447 were included and analyzed (Fig 2).

The median [IQR] time from snakebite to hospital presentation was 4h50 [2h00-17h15] ranging 0 min to 20 days. The snake involved in the bite has been brought or photographed by 183 patients (36.9%). Most of them (N = 150) were identified by the expert (Table 1). The other 33 snakes could not be identified because they had been destroyed or burned or the quality of the photo did not allow a reliable identification.

Gender-ratio (M/F) was 1.04 and the median [IQR] age was 25 [14–40] years. The distribution of patients by region, age, profession, bitten limb and season is shown in Table 2.

Prior to admission, 306 (68.5%) patients had consulted a traditional healer; treatments received are summarized in Table 3. No patient described the extraction of the snake's teeth (e.g., fangs), a fairly common practice in SSA.

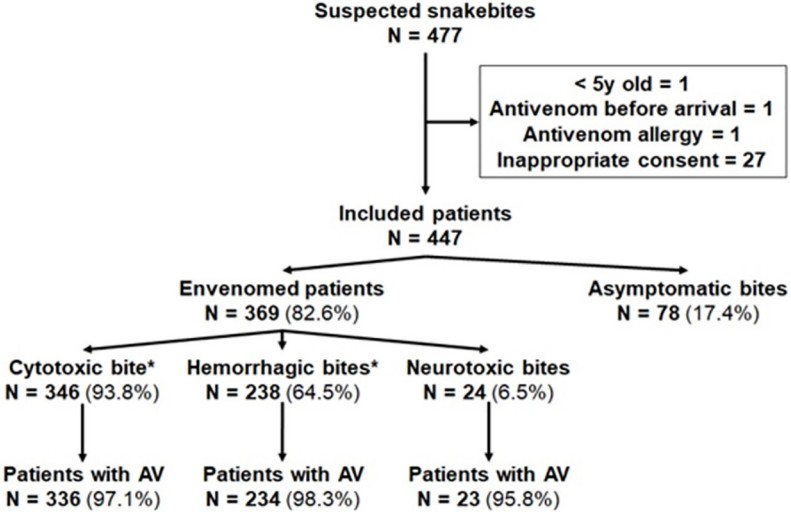

**Fig 2. Patient's clinical presentation at the time of admission.**

**Table 1. Identification of the snakes based on photos (N = 150).**

| Snake species | Number, N (%) |
|---|---|
| **Boidae** | |
| *Eryx colubrinus* | 2 (1.3) |
| **Colubridae** | |
| *Crotaphopeltis hotamboeia* | 4 (2.5) |
| *Telescopus variegatus* | 1 (0.6) |
| **Elapidae** | |
| *Dendroaspis jamesoni** | 2 (1.3) |
| *Naja haje** | 1 (0.6) |
| *Naja katiensis** | 1 (0.6) |
| *Naja melanoleuca* species complex* | 5 (3.1) |
| *Naja nigricollis** | 1 (0.6) |
| **Lamprophiidae** | |
| *Atractaspis* spp. * | 14 (8.8) |
| *Boaedon* spp. | 5 (3.1) |
| *Psammophis* spp. | 6 (3.8) |
| **Viperidae** | |
| *Atheris* spp.* | 2 (1.3) |
| *Echis romani* (formerly *ocellatus*)* | 95 (59.7) |
| *Bitis arietans** | 2 (1.3) |
| *Causus maculatus** | 8 (5) |
| **Pythonidae** | |
| *Python sebae* | 1 (0.6) |

* = Dangerous venomous snakes

Among the 447 patients enrolled, 369 (82.5%) presented SBE. Median age was 26 [14–40] years. Sex ratio (M/F) was 1.05 and 22 women were pregnant at the time of the bite. Cytotoxicity, hemorrhagic disorders and neurotoxicity was found in 346, 238 and 24 patients with SBE, respectively (Table 2). Median time (IQR) from snakebite to hospital admission was 5h44 [2h30-22h23], 6h17 [2h45-26h22], and 4h30 [1h49-21h30] in patients with cytotoxicity, hemorrhagic disorders and neurotoxicity, respectively. Among the 369 patients with SBE, 356 received at least one of whom 256 (71.9%) received one administration antivenom administration and 13 didn't (Fig 3). Among these latter, 12 presented with mild envenomation and the investigator did not consider useful to administer antivenom and one died before IPA administration.

## Patients presenting SBE and receiving at least one IPA administration

Of the 356 patients with SBE who received at least one injection of antivenom, 336 (94.4%) showed cytotoxicity, 234 (65.7%) coagulation disorders, and 23 (6.5%) neurotoxiccity (Fig 2). Total number of vials administrated according to clinical presentation at admission is presented Table 4.

One patient was considered lost to follow-up as he left the hospital, against medical advice, before the post-injection clinical evaluation was performed. Thus, 355 patients were considered to assess the efficacy of antivenom (Fig 3).

**Table 2. Patient's description.**

| | Total (N = 447) | Envenomation with IPA (N = 356) | Envenomation without IPA (N = 13) | No envenomation with IPA (N = 6) | No envenomation without IPA (N = 72) | p |
|---|---|---|---|---|---|---|
| Male gender, n (%) | 228 (51.0) | 182 (51.1) | 8 (61.5) | 6 (100.0) | 32 (44.4) | 0.049 |
| Age (years) | | | | | | 0.72* |
| Median (IQR) | 25 [14–40] | 26 [14–40] | 35 [12–49] | 30 [21–37] | 23 [13.5–35.5] | |
| Range | 5–89 | 5–87 | 5–63 | 20–47 | 5–89 | |
| Age group (years), n (%) | | | | | | 0.37 |
| 5–11 | 71 (15.9) | 58 (16.3) | 3 (23.1) | - | 10 (13.9) | |
| 12–19 | 93 (20.8) | 70 (19.7) | 2 (15.4) | - | 21 (29.2) | |
| > 19 | 283 (63.3) | 228 (64.0) | 8 (61.5) | 6 (100.0) | 41 (56.9) | |
| Region, n (%) | | | | | | 0.43 |
| North Cameroon | 322 (72.0) | 259 (72.8) | 9 (69.2) | 1 (16.7) | 53 (73.6) | |
| South Cameroon | 125 (28.0) | 97 (27.2) | 4 (30.8) | 5 (83.3) | 19 (26.4) | |
| Month, n (%) | | | | | | NE |
| January | 31 (6.9) | 21 (5.9) | 2 (15.4) | - | 8 (11.1) | |
| February | 13 (2.9) | 10 (2.8) | 1 (7.7) | 1 (16.7) | 1 (1.4) | |
| March | 40 (9.0) | 35 (9.8) | 1 (7.7) | - | 4 (5.4) | |
| April | 48 (10.7) | 38 (10.7) | 4 (7.7) | - | 9 (12.5) | |
| May | 35 (7.8) | 24 (6.7) | 1 (7.7) | 1 (16.7) | 9 (12.5) | |
| June | 34 (7.6) | 23 (6.5) | 3 (23.1) | 1 (16.7) | 7 (9.7) | |
| July | 51 (11.4) | 44 (12.4) | 1 (7.7) | - | 6 (8.3) | |
| August | 44 (9.8) | 36 (10.1) | - | - | 8 (11.1) | |
| September | 46 (10.3) | 39 (11) | - | - | 7 (9.7) | |
| October | 45 (10.1) | 35 (9.8) | 2 (15.4) | 1 (16.7) | 7 (9.7) | |
| November | 36 (8.1) | 31 (8.7) | - | 1 (16.7) | 4 (5.6) | |
| December | 24 (5.4) | 20 (5.6) | 1 (7.7) | 1 (16.7) | 2 (2.8) | |

*Kruskal-Wallis test; NE: not estimable

## Effectiveness assessment

Of the 355 patients with cytotoxicity, disappearance of cytotoxicity occurred within 2 hours in 17 (5.1%) patients, in less than 12 hours in 20 (6.0%) others and within 24 hours in 11 (3.3%) more. The probability (95% CI) of being free of edema 24h after the initial injection was 9.2% (6.4–12.9) (Fig 4). Overall, disappearance of cytotoxicity during hospitalization was observed

**Table 3. Frequency of traditional treatments reported by patients (N = 447).**

| | Yes (%) |
|---|---|
| **Traditional healer consultation** | 306 (68.5) |
| **Induction of vomiting** | 37 (8.3) |
| **Scarifications** | 153 (34.2) |
| **Black stone** | 105 (23.5) |
| **Traditional bandage** | 51 (11.4) |
| **Plant ingestion** | 103 (23.0) |
| **Tourniquet** | 151 (33.8) |
| **Sucking the bite** | 7 (1.6) |

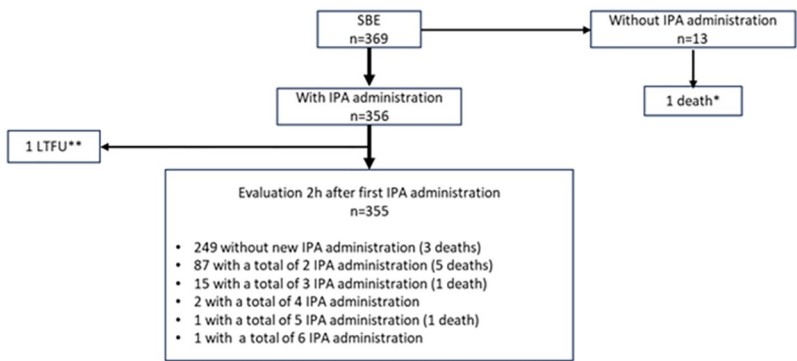

**Fig 3. Flowchart of antivenom administration in envenomed patient.** *Death before IPA administration ** left hospital against medical advice before the post-injection clinical evaluation but has been evaluated at long-term visit.

in 69 (20.6%) patients. Three days after admission, two thirds of the patients still presented edema. In multivariate analysis, delay of cytotoxicity disappearance was independently and significantly associated with the region and edema grading at admission (p = 0.001 and p<0.001, respectively). The delay to disappearance was significantly shorter in patients from South Cameroon, as compared to patients from North Cameroon. The delay to disappearance was significantly longer in patients with edema grading 2 and edema grading ≥3 at admission, as compared to patients with edema grading 1 (Table 5).

Coagulation disorders at admission was observed in 234 patients, 117 presented external bleedings and 225 had an abnormal 20WBCT (Table 4). Overall, early improvement was noted in 176 (75.2%) patients. The only two factors associated with early improvement of coagulation disorders were the region and bleeding grading at admission. In multivariate analysis, patients from South Cameroon presented lower odds of early improvement (adjusted OR [95% CI]: 0.16 [0.05–0.49]) as compared to patients from North Cameroon, and those with bleeding grade ≥3 had significantly higher odds of early improvement as compared to those with grade 1 bleeding (adjusted OR [95% CI]: 19.03 [2.16–167.47]) (Table 6). In a sensitivity analysis, only considering the 117 patients with external bleedings at enrolment, 67 (57.3%) showed early improvement. No factor was significantly associated with early improvement (S4 Appendix).

**Table 4. Number of IPA vials used during the ESAA study.**

| Patients' groups | Number of patients | Number of IPA vials | Average IPA vials (95% CI) | Median IPA vials [IQR] |
|---|---|---|---|---|
| Enrolled | 447 | 1,018 | 2.28 (2.12–2.44) | 2 [2–4] |
| Envenomed[†] | 356 | 1,003 | 2.82 (2.67–2.97) | 2 [2–4] |
| Cytotoxic ≥ grade 1[†] | 336 | 949 | 2.82 (2.67–2.98) | 2 [2–4] |
| Cytotoxic only[†] | 108 | 251 | 2.32 (2.16–2.49) | 2 [2–2] |
| bleeding ≥ grade 1 or 20WBCT ≥ stade 1[†] | 234 | 700 | 2.99 (2.79–3.9) | 2 [2–4] |
| Bleeding only ≥ grade 1 [†*] | 117* | 396 | 3.38 (3.08–3.69) | 2 [2–4] |
| 20WBCT ≥ stade 1[†] | 225 | 678 | 3.01 (2.81–3.22) | 2 [2–4] |
| 20WBCT ≥ stade 1 only[†] | 117 | 304 | 2.60 (2.35–2.85) | 2 [2–2] |
| Neurotoxic ≥ grade 1[†] | 23 | 90[$] | 3.91 (3.07–4.76) | 4 [2–4] |

IPA: Inoserp Pan-Africa; CI: confidence interval; IQR: inter quartile range

[†] with administration of IPA,

* 9 patients with bleeding and normal 20WBCT

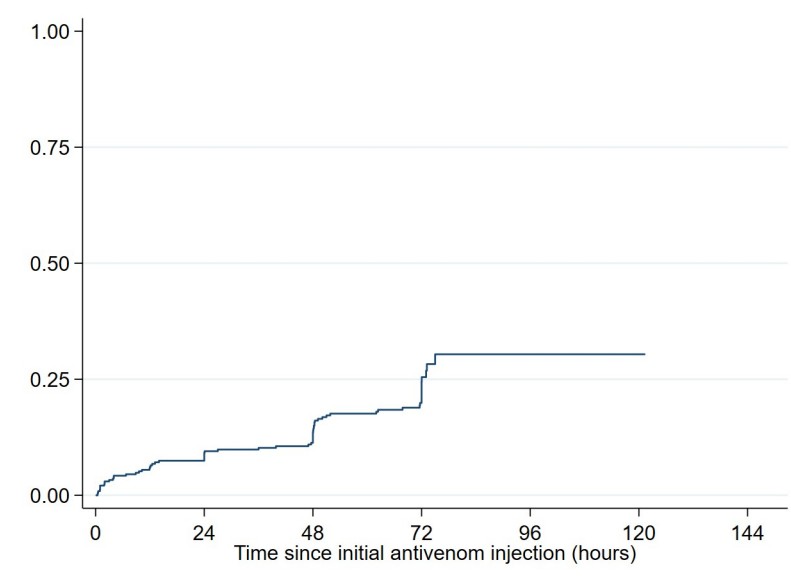

**Fig 4. Time to improvement of edema (N = 335) in patients receiving IPA.**

Among the 117 patients with external bleeding, complete stop of bleeding (grade 0) was achieved in 71 (60.7%) patients within 2 hours after first injection and in all but two patients during hospitalization (Fig 5). The probability (95% CI) to be free of bleeding 24h after the initial injection was 95.9% (91.0–98.6). The delay between snakebite and injection was the only factor significantly associated with bleeding disappearance (p = 0.005). Time to bleeding disappearance was significantly longer in those who presented 2–12 hours after SBE as compared to those who presented 12–24 hours after SBE, and it was also significantly longer in those who presented >24h after SBE (S5 Appendix).

Neurotoxicity was diagnosed in 23 patients. Early improvement was observed in 9 (39.1%) patients. Gender was the only factor associated with early improvement, female being at higher odds of early neurologic improvement (S6 Appendix). Overall, during hospitalization, neurotoxicity disappearance occurred in 19 patients. The probability (95% CI) of being free of neurological impairment 24h after the initial injection was 93.3 (74.4–99.5) (Fig 6). Age and delay from snakebite were associated with the delay of neurotoxicity disappearance in multivariate analysis (p = 0.011 and p = 0.046, respectively). Delay was significantly shorter in those aged 41 years or more as compared to those aged 20 to 40 years. The delay was significantly longer in those who presented >48 hours after SBE as compared to those who presented 2–12 hours after SBE (S7 Appendix).

## Home visit evaluation

At hospital discharge, 27 patients showed local cytotoxic complications that were not yet cured. A home visit was carried-out in 302 of the 356 (84.8%) patients with SBE and who received IPA in median (IQR) 15 [14–20] days after hospital admission. We were able to confirm 9 aesthetic (unsightly scars) and/or functional (motor deficit, amputation, deformity or ankylosis) sequelae (3.0%). The two amputations observed were minimal i.e., involving only the finger or toe (Table 7). Among patients with local lesion at the time of hospital discharge,

**Table 5. Factors associated with disappearance of cytotoxicity signs (AFT, N = 335).**

| | Crude TR (CI 95%) | P | Adj. TR (CI 95%) | P |
|---|---|---|---|---|
| Gender | | 0.79 | | |
| Male | 1 | | | |
| Female | 1.10 (0.54–2.25) | | | |
| Age (in years) | | 0.55 | | |
| 5–11 | 1.62 (0.58–4.48) | | | |
| 12–19 | 0.98 (0.38–2.56) | | | |
| 20–40 | 1 | | | |
| >40 | 1.78 (0.68–4.69) | | | |
| Time since snakebite | | 0.014 | | 0.07 |
| [0-2h] | 0.09 (0.02–0.41) | | 0.50 (0.21–1.18) | |
| [2h-12h] | 0.36 (0.10–1.25) | | 1 | |
| [12h-24h] | 1 | | 1.00 (0.35–2.90) | |
| [24h-48h] | 0.52 (0.11–2.43) | | 0.49 (0.18–1.32) | |
| $\geq$ 48H | 0.22 (0.05–0.92) | | 0.34 (0.15–0.78) | |
| Traditional medicine | | 0.69 | | |
| Yes | 1 | | | |
| No | 1.17 (0.54–2.55) | | | |
| Treatment before arriving at the center | | 0.27 | | |
| Yes | 1 | | | |
| No | 1.60 (0.69–3.75) | | | |
| Region | | <0.001 | | 0.001 |
| North Cameroon | 1 | | 1 | |
| South Cameroon | 0.22 (0.11–0.45) | | 0.35 (0.19–0.67) | |
| Glasgow score at admission | | 0.63 | | |
| < 15 | 2.01 (0.11–37.83) | | | |
| 15 | 1 | | | |
| Edema before injection | | <0.001 | | <0.001 |
| 0–1 | 1 | | 1 | |
| 2 | 9.09 (4.38–18.86) | | 7.42 (3.67–14.99) | |
| $\geq$3 | 30.56 (9.01–103.68) | | 21.01 (6.75–65.44) | |
| Hemotoxicity before injection | | <0.001 | | |
| 0 | 0.21 (0.10–0.47) | | | |
| 1 | 0.17 (0.06–0.49) | | | |
| 2 | 1 | | | |
| $\geq$3 | 0.92 (0.22–3.83) | | | |
| Neurotoxicity before injection | | 0.10 | | |
| 0–1 | 1 | | | |
| $\geq$2 | 0.27 (0.06–1.24) | | | |

21 complete recoveries were confirmed and 6 patients could not be traced, and it was not possible to assess the evolution of local lesions.

## Clinical histories of deceased patients

Among the 369 patients with SBE, 11 (3%) died, 2 from elapid envenomation with neurotoxic symptoms, including one patient who died before IPA administration (patient 1), 7 showing bleeding from the snakebite (all by *E. romani*), 1 from sepsis-like syndrome after *N. nigricollis*

**Table 6. Baseline factors associated with early improvement of coagulation disorders (logistic regression, N = 234).**

| | n | Early improvement (%) | Crude OR (95% CI) | p | Adj. OR (95% CI) | p |
|---|---|---|---|---|---|---|
| Gender | | | | 0.59 | | |
| Male | 126 | 93 (73.8) | 1 | | | |
| Female | 108 | 83 (76.8) | 1.18 [0.65–2.14] | | | |
| Age (in years) | | | | 0.50 | | |
| 5–11 | 40 | 29 (72.5) | 0.70 [0.30–1.63] | | | |
| 12–19 | 44 | 34 (77.3) | 0.90 [0.38–2.12] | | | |
| 20–40 | 100 | 79 (79.0) | 1 | | | |
| > 40 | 50 | 34 (68.0) | 0.56 [0.26–1.21] | | | |
| BMI (> 20 years) (N = 141) | | | | 0.12 | | |
| < 18.5 | 11 | 7 (63.6) | 0.37 [0.10–1.44] | | | |
| 18.5–24.9 | 80 | 66 (82.5) | 1 | | | |
| 25–39,9 | 26 | 16 (61.5) | 0.34 [0.13–0.90] | | | |
| ≥ 40 | 24 | 17 (70.8) | 0.52 [0.18–1.48] | | | |
| Time since snakebite | | | | 0.82 | | |
| [0-2h] | 22 | 18 (81.8) | 1.59 [0.50–5.09] | | | |
| [2h-12h] | 111 | 82 (73.9) | 1 | | | |
| [12h-24h] | | 25 (80.6) | 1.47 [0.55–3.95] | | | |
| [24h-48h] | 31 | 24 (75.0) | 1.06 [0.43–2.62] | | | |
| ≥ 48H | 3238 | 27 (71.0) | 0.86 [0.38–0.97] | | | |
| Traditional medicine | | | | 0.29 | | |
| No | 64 | 45 (70.3) | 1 | | | |
| Yes | 170 | 131 (77.1) | 1.42 [0.74–2.71] | | | |
| Treatment before arriving at the center | | | | 0.37 | | |
| No | 172 | 132 (76.7) | 1 | | | |
| Yes | 62 | 44 (71.0) | 0.74 [0.39–1.42] | | | |
| Region | | | | 0.005 | | 0.001 |
| North Cameroon | 210 | 164 (78.1) | 1 | | 1 | |
| South-Cameroon | 24 | 12 (50.0) | 0.28 [0.12–0.67] | | 0.16 [0.05–0.49] | |
| Glasgow score at admission | | | | 0.80 | | |
| <15 | 5 | 4 (80.0) | 1.32 [0.15–12.10] | | | |
| 15 | 229 | 172 (75.1) | 1 | | | |
| Edema grading at admission | | | | 0.71 | | |
| 0–1 | 74 | 58 (78.4) | 1 | | | |
| 2 | 96 | 70 (72.9) | 0.74 [0.36–1.52] | | | |
| ≥3 | 64 | 48 (75.0) | 0.83 [0.38–1.83] | | | |
| Bleeding grading at admission | | | | 0.003 | | 0.028 |
| 1 | 44 | 28 (63.6) | 1 | | 1 | |
| 2 | 164 | 123 (75.0) | 1.71 [0.84–3.48] | | 1.06 [0.47–2.40] | |
| ≥3 | 26 | 25 (96.1) | 14.28 [1.77–115.61] | | 19.03 [2.16–167.47] | |
| Neurotoxicity grading at admission | | | | 0.56 | | |
| 0–1 | 225 | 170 (75.6) | 1 | | | |
| ≥2 | 9 | 6 (66.7) | 0.65 [0.16–2.67] | | | |

OR: odds ratio; CI: confidence interval; BMI: body mass index

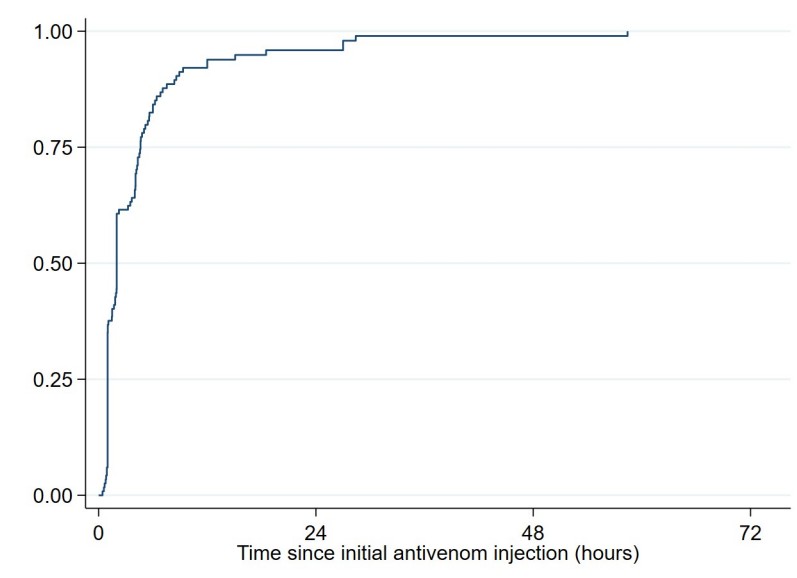

**Fig 5. Time to stop bleeding (N = 117) in patients receiving IPA.**

bite and 1 probably not bitten by a snake but poisoned by plants used for traditional treatment (Table 8).

Patient 2 was probably not bitten by a snake. On arrival at the hospital 3 hours after the event, the patient (who did not see what injured him) was agitated and presented without pain, no local edema or bruising, normal WBCT, without ptosis, non-bloody vomiting, profuse diarrhea without mucus or blood and severe epigastric pain. Eight hours after the event,

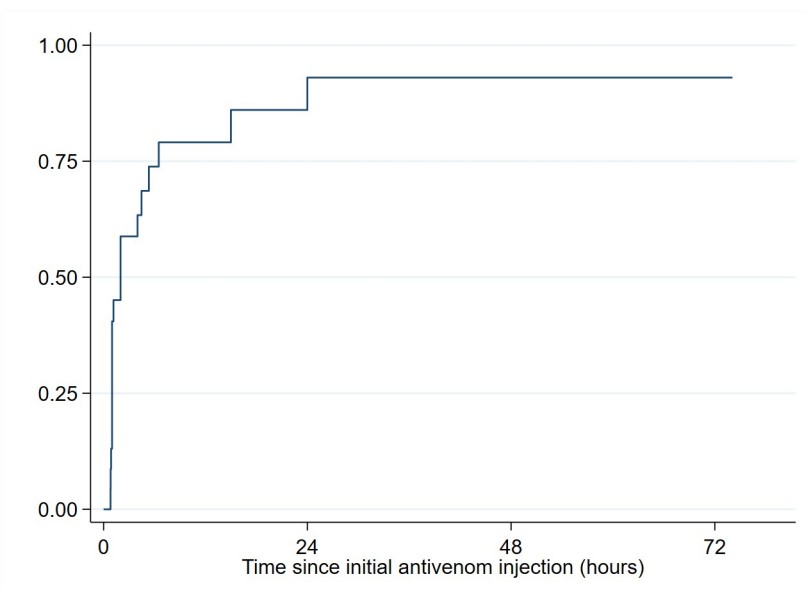

**Fig 6. Time to stop of neurotoxic signs (N = 23) in patients receiving IPA.**

**Table 7. Sequalae observed in 9 patients during home visit.**

| Locality | Snake | Time to treat | Number of vials | Sequelae |
|---|---|---|---|---|
| Kolofata | *Echis romani* | 26h15 | 4 | Thumb necrosis |
| Tokombéré | *Echis romani* | 3h15 | 4 | Deformity and ankylosis of the index finger |
| Tokombéré | ? | 3 days | | Skin necrosis scar |
| Tokombéré | *Echis romani* | 16 days | 0 | Middle finger amputation |
| Guider | *Echis romani* | 4h20 | 2 | Ring finger amputation |
| Guider | *Echis romani*\* | 26h30 | 2 | Deformity and major ankylosis |
| Guider | ? | | 2 | Skin necrosis scar |
| Guider | *Echis romani*\* | 2 days | 6 | Skin necrosis scar |
| Yaoundé | *Bitis nasicornis*\* | 9h25 | 6 | Skin necrosis scar |

\* Snake not brought but identified from the poster

he showed isolated polypnea. Seventeen hours after the event, small amounts of blood appear in the vomitus. The patient died 22 hours after the event. Envenomation (and possibly snakebite) was excluded and the diagnoses retained, after expert opinions, were pulmonary embolism, viral disease (including SARS-CoV2 or Ebola), gastroenteritis, peptic ulcer or other gastrointestinal hemorrhage, and poisoning by a plant contained in the traditional treatment administered before admission to hospital.

A fourth 80-year-old patient (patient 7) died after being bitten by *N. nigricollis*. He had a known chronic heart failure prior to the bite and showed no bleeding or neurotoxic symptoms. The death could be related to his cardiac history and/or sepsis.

Patient 9 bitten by *N. haje* did not show neurological disorders (i.e., ptosis or dyspnea) at any time, but significant local envenomation (edema and blisters) that increased during the

**Table 8. Deceased patients.**

| Patient number | Snake | Locality | Place of death | Age | Time to treatment | Time to death | AV dose | Link with SAV | Cause of death |
|---|---|---|---|---|---|---|---|---|---|
| 1 | Elapidae? | Tokombéré | Hospital | 6 | 1 h 30 | 2 h | 0 vial | No | Respiratory failure |
| 2 | Doubtful snakebite | Akonolinga | Hospital | 24 | 5 h | 22h30 | 6 vials | No | Digestive hemorrhage or plant poisoning (traditional treatment) |
| 3 | *Echis romani*[§] | Guider | Hospital | 12 | 1 or 2 h | 6/7 h | 4 vials | No | Hemorrhagic syndrome + malaria + insufficient antivenom dose |
| 4 | *Echis romani*[&] | Mora | Hospital | 20 | 158 h | 165 h | 4 vials | No | Severe anemia + brain hemorrhage |
| 5 | *Echis romani*[§] | Poli | Home\* | 45 | 3 h | 168 h | 6 vials | No | Anemia + malaria |
| 6 | *Echis romani*[&] | Poli | Home\* | 41 | 7 h | 120 h | 4 vials | No | Anemia |
| 7 | *Naja nigricollis*[§] | Tokombéré | Home\* | 80 | 4 h | 500 h | 2 vials | No | Heart failure unrelated to envenomation + envenomation |
| 8 | *Echis romani*[&] | Poli | Hospital | 42 | 24 h | 117 h | 4 vials | No | Cardiovascular Collapse |
| 9 | *Naja haje*[§] | Tokombéré | Hospital | 8 | 17.4 h | 49.6 h | 4 vials | No | Sepsis, inhalation of vomiting, intoxication from traditional treatment, or pulmonary embolism |
| 10 | *Echis romani*[§] | Poli | Hospital (Garoua) | 25 | 1.7 h | 114.3 h | 10 vials | No | Acute renal failure + obstetric hemorrhage due to envenomation |
| 11 | *Echis romani*[§] | Poli | Hospital | 8 | 62.5 h | 63.5 h | 2 vials | No | Brain hemorrhage |

[§] = snake brought by the patient;

[&] = snake identified from the poster;

\* = discharged against medical advice

course of envenomation. She died 50 hours after the bite showing respiratory distress followed by cardiac arrest. The absence of any sign of intolerance following the first administration of antivenom and the onset of serious clinical disorders, both local and systemic, 34 hours after admission when the patient seemed to improve, suggest a sepsis, inhalation of vomiting, intoxication from traditional treatment, or pulmonary embolism rather than direct envenomation consequences.

Out of 22 enrolled pregnant women who were envenomed, one lost her baby (estimated pregnancy 36 weeks). She arrived at hospital 23 hours after *E. romani* bite and had grade 3 edema, no bleeding and an abnormal WBCT treated with 2 vials of IPA. Fetal's ultrasound performed 38 hours after the bite revealed intrauterine fetal demise. Less than 6 days after the bite, the mother expulsed a 3rd degree macerated baby consistent with an in-utero death dating back 8–12 days. In addition, the pregnancy follow-up file showed fetal growth arrest and maternal weight loss of approximately 2 kg during the month preceding the bite.

## Discussion

Our study, performed in real-life condition, without dedicated study health personal or specific logistical material, allowed to describe population bitten by snake in this area, healthcare referring practices and the effectiveness of an antivenom serum administration.

As in most clinical studies carried out in SSA, the time between the bite and hospital presentation was very long for a majority of patients [3,27]. This was explained by a complex treatment seeking behavior favoring traditional medicine, and reluctance of people facing the poor accessibility of antivenoms [5,14,27–29]. In addition, some patients in this study waited for the onset of complications, in particular cerebral bleeding or local worsening, to come to the hospital. This could result in a significant loss of effectiveness of the antivenom whose role is to eliminate the venom before the onset of complications rather than to directly treat the symptoms. Regarding Elapidae neurotoxic SBEs, the delay further reduces the effectiveness of IPA due to early binding of neurotoxins to cholinergic receptor.

The patients enrolled in the study were mostly young people bitten during agricultural and pastoral activities, as is generally the case in SSA. The epidemiological characteristics, in particular the delay between the bite and hospital presentation, and the proportion of patients having one of the three main syndromes, did not differ from those observed in most studies carried out in SSA, with the exception of the gender ratio, that was at equilibrium in this study, whereas there is usually a significantly higher number of males [3,12,18,22,27]. The patients enrolled here appears to be representative of current situations in SSA, except for particular places such as Guinea where the incidence of Elapidae bites is high, which is perhaps also observed in the Congolese forest block including southern Cameroon [3,12,30].

In this study, we managed to identify the species of more than a third of the snakes responsible for SBE, which is rarely achieved in clinical studies in SSA. The identification carried out thanks to the photos of the snakes brought by the victim is simple, precise and accurate. On the other hand, species recognition carried out from the poster proved to be unreliable. In many cases, snake identified by the patient was inconsistent with geographic region (despite the distribution map associated to the snake picture) or clinical symptomatology. A better identification could perhaps be obtained by limiting the species presented on the poster to those actually present in the geographical area of the health center.

The study did not identify factors associated with early improvement or worsening of IPA-treated SBEs except region and hemotoxic grade before injection considering coagulation disorders, and gender for neurotoxic envenomation. Regarding coagulation disorders, multivariate analysis showed a borderline effect of time to treatment in term of early improvement in

the sensitive analysis considering only patients with external bleeding. Time to treatment was significantly associated when we considered factors associated with complete stop bleeding. Regarding impact of the region where snakebite occurred, one explanation could lie in the species, which differ from region to region. In northern Cameroon, *Echis romani* is the most common species, whereas it is totally absent from southern Cameroon, where it is replaced by *Bitis* spp. On the one hand, the metalloproteinases in *Echis* venom act on the vascular endothelium, leading to hemorrhage, and on the other, the prothrombin activator leads to consumption of blood factors, resulting in blood that is incoagulable (see below) [31,32]. On the other hand, *Bitis* venom contains a thrombin-like enzyme that induces fibrinogen consumption, running to incoagulable blood [31,33]. However, the uncertainties and approximations associated with clinical grading must also be taken into account, particularly when there are multiple graders, which can bias the interpretation of patient improvement. Grade reduction is undoubtedly more frequent for higher than for lower grades. Curiously, female gender was associated with higher odds of early improvement in patients who presented neurotoxic signs. The effect of gender has not been previously reported in the literature, and we did not see clear explanation to this association. However, solely 23 patients showed neurotoxic symptoms, making it impossible to search for significant associations.

On the other hand, the study showed that the administration of IPA resulted in a good improvement in the hematological and neurologic symptoms of envenomation. Within 2 hours of first IPA administration, 2 vials stopped bleeding in 71 patients (60.7%). Overall, 95.9% of patients with external bleeding and 93.3% of patient with neurotoxic signs were free of bleeding and free of neurological impairment, 24h after the initial injection respectively.

The study also confirms that IPA, no more than other antivenoms, is unable to stop or even slow the progression of edema, which lasts for several days after the start of treatment, even at high doses. Edema is an early criterion for viper envenomation, the most common, but a poor indicator of clinical progression, even after administration of antivenom [18,22].

Despite the observed effectiveness of IPA, attention should be paid to the outcomes concerning severe SBEs, in particular sequelae or complications in pregnant patients, and death.

The frequency of sequelae was low and were mostly mild, which confirms most recent studies in this region [5,14–17]. These results seem to be independent of the effectiveness of the IPA, especially since the time between the bite and presentation to the hospital, as well as the number of vials administered are highly variable.

Deaths, representing 3% of envenomed patients and 2.8% of patients treated with IPA. Our results were similar to those reported in other studies on IPA or FAV Afrique [14,15,18,34]. Surprisingly, of the 9 envenomation-related deaths treated with IPA, no patient died of respiratory paralysis related to Elapidae neurotoxic envenomation.

However, the death of 7 patients in association with coagulation disorders raises the question of the cause of death and adequate dose of IPA. Death from bleeding disorders can result from several causes that are not always identifiable.

At the beginning, venom-induced consumptive coagulopathy (VICC) results from activation of coagulation by snake venom enzymes, including thrombin-like enzymes, prothrombin activators, and factor X activators [31]. The blood becomes incoagulable, which causes abnormal WBCT and bleeding in 62% and 32% of patients, respectively. The onset and resolution of VICC are rapid, especially if antivenom administration is early. In some patients, VICC progresses to thrombotic microangiopathy characterized by acute renal failure, thrombocytopenia, and microangiopathic hemolytic anemia (MAHA). The latter continues despite resolution of the coagulopathy, suggesting a separate but related process [31]. The existence of delayed hyperfibrinolysis has been described in envenomation by *E. pyramidum*, a species from East

Africa close to *E. romani*. The action of antivenom is rapidly favorable on VICC but probably less on delayed hyperfibrinolysis [35,36].

Here, the study makes it possible to describe three clinical situations reflecting the instability of coagulation in addition to the delay between snakebite and antivenom injection significantly associated with bleeding disappearance.

- In some cases, it was not possible to exclude that the dose of IPA was insufficient, as for patient 3 (4 vials which were enough to stop the exteriorized bleeding) who presented neurological disorders wrongly attributed to an Elapidae bite when it was a probable early cerebral hemorrhage on which no antivenom can act. This might also be the case for patient 8 who died despite the cessation of bleeding observed after the second administration of IPA, and of the pregnant woman (patient 10) who died from postpartum hemorrhage, when the bleeding had initially stopped and the baby was delivered healthy.

- Some patients arrived late at the hospital and had complications between the bite and the arrival at the hospital, which is often the reason for consultation. These are severe anemia or internal bleeding for which antivenom is poorly effective in the absence of substitutive or supportive treatment, as for patients 4 and 6.

- For others, failure to stop bleeding or recurrence of bleeding when the treatment seemed effective (patient 5, 6 and 10) may result from complex clotting disorders like VICC followed by delayed hyperfibrinolysis [32].

Coagulation disorders are responsible for the majority of deaths in SSA health centers. Despite reduced case fatality rate compared to no antivenom treatment, coagulation disorders result from a succession of clinical presentations the causes of which seem to become autonomous during the progression of the envenomation. The measurement of venonemia (plasma concentration of venom) would allow to better understand this phenomenon and will be carried on in future studies. However, this illustrates the strongly need to combine antivenom with symptomatic treatment needed to either supplementing the defective coagulation factors or stopping the deleterious process at the origin of the bleeding complications.

The study had some limitations. It was strongly impacted by the security situation in the NC–which had been anticipated during the preparation–and the Covid-19 pandemic which forced us to adapt both organizationally and logistically. The follow-up of WBCT20 after H2 was irregular, which does not allow us to use this variable in the analysis of the effectiveness of the IPA. In addition, the D15 visit was missing in 16.6% of patients who received IPA and many of them were done with a delay of several weeks, sometimes several months. However, this delay benefits the evaluation of sequelae which could be described more precisely than generally performed in clinical studies.

Nevertheless, there were very few patients lost to follow-up–only 3 (0.8%) patients treated with IPA–during the duration of the hospitalization. The patient management algorithm was respected in more than 95% of the participants.

## Conclusion

The ESAA study involving 447 patients, including 356 envenomed patients treated with IPA, showed that the administration of a minimum dose (2 or 4 vials depending on the clinical presentation) was sufficient to obtain an early improvement in envenomation in most of them. By ensuring a strict monitoring within 2 hours of administration of a dose of IPA corresponding to 500 $LD_{50}$ for Viperidae (edema, and/or bleeding and/or abnormal WBCT) or 1,000 $LD_{50}$ for Elapidae (signs of neurotoxicity such as ptosis and/or dyspnea), it is possible to improve

the clinical course of SBE. The renewal of the same dose of IPA two hours later must be carried out in the event of persistence of bleeding or neurotoxic symptoms. A clinical reassessment every 2 hours is essential to ensure patient safety while preserving the antivenom resource. 20WBCT can't be use as criteria of antivenom re-administration due to poor specificity and long delay before normalization [31,37,38]. The clinical and especially biological criteria should be further refined by other studies which must also identify effective symptomatic treatments to treat coagulation instability.

In addition to the effectiveness of the IPA administered according to the sequential administration strategy, the ESAA study showed its excellent tolerance, which will be the subject of a specific publication.

Ad hoc training of healthcare personnel in the clinical diagnosis of envenomation, correct administration of antivenom, and monitoring of treatment is essential to reduce disability and mortality from SBE.

## Supporting information

**S1 Appendix. Management algorithm recommended by Cameroonian Ministry of envenomation patients.**
(DOCX)

**S2 Appendix. Gradation of whole-blood clotting time on dry tube (WBCT).**
(DOCX)

**S3 Appendix. List of snakes for which the AV is effective (in bold, snake species present in Cameroon).**
(DOCX)

**S4 Appendix. Baseline factors associated with early improvement of coagulation disorders (logistic regression, N = 117).**
(DOCX)

**S5 Appendix. Factors associated with complete stop bleeding (AFT model, N = 117).**
(DOCX)

**S6 Appendix. Baseline factors associated with early improvement of neurotoxicity (logistic regression, N = 23).**
(DOCX)

**S7 Appendix. Factors associated with improvement of neurotoxic signs (AFT, N = 23).**
(DOCX)

## Acknowledgments

We thank the field investigators without whom this work would not have been possible: Dr Metogo et M. Boris Kouomogne (CURY, Yaoundé), Dr Larissa Mvogo (Akonolinga), Dr Foe et Madame Fouda (Sa'a), Dr Guillaume Gayma et Dr Fadimatou (Mora), Dr Christophe Youmbi et M. Kouli Guidang (Kolofata), Dr Baldagai et Dr Olivier Bito (HR Ngaoundéré), Dr Hans Notaya (HN Ngaoundéré), Dr Hamdja Moustafa (Poli), Dr Ousmana et Mme Ninkouague Jogo (Guider), Dr Mokake (Buea), Dr Njie Thompson Kingue et Dr George Ethe Ethe (Limbe), Dr Emilenne Takeng (Tiko), Dr Metomo et Mme Bertha EOCK (Njombe Penja). We would like to thank all the patients who accepted to participate in this study, as well as the regional chief medical officers and the medical staff who participated in this study. We are grateful to Inosan Biopharma for providing the antivenoms free of charge. We are indebted to

Prof. Leslie Boyer and Dr Ellen Einterz for their valuable comments on cases, in particular the serious clinical events, and participation in the scientific committee. We also thank Dr. Luc de Haro, centre antipoison de Marseille, France, for his relevant expertise.

## Author Contributions

**Conceptualization:** Jean-Philippe Chippaux, Yap Boum, Armand S. Nkwescheu, Fabien Taieb.

**Data curation:** Rodrigue Ntone, Gaëlle Noël, Fai Karl, Pierre Amta, Lucrece Matchim, Pedro Clauteaux, Lucrèce Eteki, Mark Ndifon.

**Formal analysis:** David Benhammou, Yoann Madec.

**Funding acquisition:** Fabien Taieb.

**Investigation:** Pierre Amta.

**Methodology:** Jean-Philippe Chippaux, Fabien Taieb.

**Project administration:** Rodrigue Ntone, Gaëlle Noël, Anais Perilhou, Pedro Clauteaux, Yap Boum, Armand S. Nkwescheu.

**Resources:** Rodrigue Ntone, Anais Perilhou.

**Software:** Marie Sanchez.

**Supervision:** Jean-Philippe Chippaux, Rodrigue Ntone, Gaëlle Noël, Fai Karl, Pierre Amta, Lucrece Matchim, Pedro Clauteaux, Lucrèce Eteki, Mark Ndifon, Yap Boum, Armand S. Nkwescheu, Fabien Taieb.

**Validation:** Jean-Philippe Chippaux, Rodrigue Ntone, Fai Karl, Pierre Amta, Lucrece Matchim, Lucrèce Eteki, Mark Ndifon, Yap Boum, Armand S. Nkwescheu, Fabien Taieb.

**Visualization:** Jean-Philippe Chippaux, David Benhammou, Yoann Madec, Gaëlle Noël, Fabien Taieb.

**Writing – original draft:** Jean-Philippe Chippaux, David Benhammou.

**Writing – review & editing:** Jean-Philippe Chippaux, Yoann Madec, Gaëlle Noël, Yap Boum, Armand S. Nkwescheu, Fabien Taieb.

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
