## [Decision Letter · Decision Letter 0]

24 Mar 2023

Dear Professor Chippaux,

Thank you very much for submitting your manuscript "Field evaluation of InoserpTM PAN-AFRICA antivenom effectiveness in Cameroon" for consideration at PLOS Neglected Tropical Diseases. As with all papers reviewed by the journal, your manuscript was reviewed by members of the editorial board and by several independent reviewers. In light of the reviews (below this email), we would like to invite the resubmission of a significantly-revised version that takes into account the reviewers' comments. 

Please pay particular attention to the comments from reviewer 1, who has provided a detailed review and advised various important modifications. I look forward to seeing the revised article in due course.

Kind Regards,

Dr Michael Abouyannis

We cannot make any decision about publication until we have seen the revised manuscript and your response to the reviewers' comments. Your revised manuscript is also likely to be sent to reviewers for further evaluation.

Sincerely,

Michael Abouyannis

Guest Editor

Wuelton Monteiro

Section Editor

Thank you for submitting your paper to PLOS NTD. Considering the reviewers' comments, you are invited to make major revisions to the article. Please pay particular attention to the comments from reviewer 1, who has provided a detailed review and advised various important modifications. I look forward to seeing the revised article in due course.

Kind Regards,

Dr Michael Abouyannis

Reviewer's Responses to Questions

**Key Review Criteria Required for Acceptance?**

**Methods**

-Are the objectives of the study clearly articulated with a clear testable hypothesis stated?

-Is the study design appropriate to address the stated objectives?

-Is the population clearly described and appropriate for the hypothesis being tested?

-Is the sample size sufficient to ensure adequate power to address the hypothesis being tested?

-Were correct statistical analysis used to support conclusions?

-Are there concerns about ethical or regulatory requirements being met?

Reviewer #1: The objectives are not completely clear and I have some concerns about the study design/sample size and the ethics (see text below)

Reviewer #2: Excellent study design given the location in Western Africa for the assessment. Targeting regions with different habitation of venomous snakes of concern is important and was assessed. Population was defined and inclusion and exclusion criteria are reasonable for this investigation. Sample size is also calculated for sufficient power but as author's mentioned, they did not reach their goal, but it did not have an impact on statistical significance. 

Some comments and questions to address: 

 - If patient did not consent but was deemed eligible for study (envenomated) then what was the protocol for clinical care? Please add some details in this situation. Looking at Figure 2, it reveals that 27 individuals were deemed "inappropriate" consent and were excluded. What were the circumstances of why the consenting process was deemed inappropriate. This should be included in the limitations and also in the methods section. Recommend explaining in more detail these two points.

Reviewer #3: Line 204 - 207. The additional dose supplied is not described in the text

**Results**

-Does the analysis presented match the analysis plan?

-Are the results clearly and completely presented?

-Are the figures (Tables, Images) of sufficient quality for clarity?

Reviewer #1: the analysis does not seem to match the objectives. The tables/images/figures are so many that it makes it difficult for the reader to keep track.

Reviewer #2: Results are sufficient for analysis. They are presented clearly aside from comment below. Tables and figures are sufficient. 

- Line 308 - 311 and table 9, Lines 430 439. I am curious why the individual who was "probably not bitten by a snake" was included in study. Please explain further as it states, "envenomation was excluded". In the clinical history provided it appears to have been reasonable to administer IPA initially but as the case evolved it sounds as if another etiology was at hand. In my opinion, an adjustment may be needed to the mortality calculations to reflect 10 participants. The reason why this is so important is that at least 10 patients did die even after being given IPA via protocol being used. These individuals likely may have needed more vials of antivenom, but this is hard to know at this time. Authors do mention that plasma concentrations of venom were not done and being planned for future study.

Reviewer #3: Line 253. 24 unidentified snakes are presented, while in table 2 there are 10

The text does not show data on blood coagulation values that would allow evidence of its correction in the two hours following the administration of the antivenom.

**Conclusions**

-Are the conclusions supported by the data presented?

-Are the limitations of analysis clearly described?

-Do the authors discuss how these data can be helpful to advance our understanding of the topic under study?

-Is public health relevance addressed?

Reviewer #1: I wonder if the conclusions can be reorganised and then follow the objectives/results in an easier to read text.

Reviewer #2: Conclusions are reasonable and discussed. Each death was also discussed and important. Discussion is thorough and draws on relevant literature. Limitations were discussed and reasonable.

Reviewer #3: No comment

**Editorial and Data Presentation Modifications?**

Reviewer #1: (No Response)

Reviewer #2: I recommend accept with minor revision.

Reviewer #3: I suggest that evidence of blood coagulation values be shown. This will make it possible to demonstrate its correction in the two hours following the administration of the antivenom.

**Summary and General Comments**

Reviewer #1: The paper describes the clinical outcomes of a snakebite patient cohort treated with Inoserp antivenom. Data on clinical outcomes are highly needed. But I do have some concerns about this study as it is currently not completely clear which research question the authors are trying to answer and how exactly. I hope the authors can clarify this by reorganizing the data such that the readers can understand the flow better; focusing on their main questions, pulling in the relevant data from the tables/figures in the attachments and remove unneeded tables and analyses irrelevant to the main questions. 

In more detail: 

-It is not clear which components of the clinical follow-up and therapeutic algorithm in the cohort was part of the standard care in the hospitals involved. Which is relevant to know when interpreting the data. 

-In the introduction it is stated that patients normally have to pay for their antivenom, but in the methods it is explained that antivenom and symptomatic treatment were provided free of charge to the patients in the study. This puts pressure on patients to sign the consent form. Or was antivenom and symptomatic treatment also free for patients who refused to participate? What did the Ethics committees ask about this access to treatment? And how long after the cohort study did thing return to normal (pay for antivenom)? How did that impact health seeking behavior as they were expecting free treatment? 

-It is unclear what the main research question was. It switches between effectiveness and tolerability. But effectiveness is not well defined as it’s a cohort. Which is probably why the authors picked adverse effects as the outcome measure used in the sample size calculation. But if that outcome measure adverse effect is used to calculate sample size this is the primary outcome but it’s hardly discussed in the results and discussion and is not in the abstract. Sample size calculation therefore is not in line with the rest of the manuscript. 

How to define effectiveness of Avs if it’s unknown what would have happened without AV or with an other AV; why not rephrase the manuscript describing outcomes of a patient cohort when AV is used. 

‘Factors associated with early effectiveness were identified using logistic regression models. Didn’t the authors look at factors associated with less severe disease instead of early effectiveness? How to distinguish these two in the current data available? Which variables were tested for effect modification apart from looking at confounders only? 

Please use the https://www.strobe-statement.org/checklists/ to report methodology and results from a cohort.

The ethics paragraph states there were scientific committees to discuss the results. A representative of the sponsor (Institut Pasteur, Paris) was attending apart from the main investigators from Paris. Who was this representative? As it is also explained that Inoserp financially supports Institut Pasteur, Paris (financial statement: We are grateful to Inosan for providing the antivenoms free of charge and partially funding the study through a partnership with Institut Pasteur, Paris, sponsor of the study). Can the authors explain this partnership? This should probably be reported in the conflict of interest?

I am confused by the different numbers. Eg In the results there are 336 cytotoxic symptoms with at least one antivenom injection, in the abstract in 346 and in the table 5 cytotoxic only 108 + cytotoxic and bleeding 234 is together 342. Do the authors think the model they developed to look at effectiveness works for the 108 patients with cytotoxic problems? 

I have challenges reading the discussion; can authors remove info on patient outcomes/case histories to the results and move p=values (or preferably effect measures) to the results? Referring to tables in the discussions makes it also more difficult for the reader to follow the main messages. 

Title: Why is there a ‘field evaluation’ mentioned in the title whereas it is all about hospital treatment?

Abstract: 

Line 26 However: what is the contrast? And mentioned is effectiveness urgently needed and then the next line ‘ we assessed tolerance and effectiveness’. Not clear; is your main goal tolerance or effectiveness?

Study design is not included (cohort study). Explain standard practice versus procedures followed in cohort. 

Introduction:

Line 100-101 ‘the main problem encountered in SSA is the poor accessibility to antivenoms’. That will depend per country. In many countries/regions the availability is worse than the accessibility. 

Line 109 ASS (French)

Some more data from ref 14 and 15 on Inoserp is needed which can also help us to understand which data is missing from these cohorts which will be obtained in the current study. 

Authors state that ‘patients refuse to buy more than one or two vials at the time’. That cannot be generalized for entire SSA. 

Methods: clinical study -> cohort

Why are patients younger than 5 years excluded? 

Line 186 E ocellatus -> romani

Assessment of clinical outcomes: what if patients initially improve but then deteriorate? 

Statistical analysis: all 356 -> number of patients included in the methods should move to the results instead of being in the methods section. 

Please discuss the validity of the definition of ‘early effectiveness/medium-term effectiveness’ model used. 

Line 230: 'some ad hoc analyses' is too vague. And there are many ad hoc analyses in the appendices; can you reduce the number of analyses based on the research questions to avoid unexpected, biologically difficult to explain p-values?

Ethics: please add the numbers of the ethics evaluations. 

Results line 154; please provide RR + intervals instead of p-values. 

Lines 264-266: figure briefly describes clinical presentation of the patients, including death? Now reads as if patients came in dead.

The first line in the results/effectiveness assessment is on the storage conditions of IPA instead of a topic related to effectiveness?

Line 287: ‘Improvement was noted within 2 hours after the first antivenom administration in 189 patients. This means a positive 20WBCT switched to negative? No INR data? How many missing 20WBCT do you have (as reported in the discussion as a limitation)? How many had a positive 20WBCT, then negative and then back to positive? 

Line 289: ‘ complete stop of bleeding was achieved within 2 hours’. How was this defined? As it can be quite difficult to know for sure a bleeding stopped within 2 hours? Minor or major bleedings? 

Line 299: Multivariate analysis time between snakebite and antivenom injection was significantly associated with early improvement… and then a p-value follows instead of an effect size (OR + interval). Please report results in line with multivariate analysis instead. Another question on this analysis is; The cytotoxic patients presented later and it takes more time to recover. So how do you know you aren’t finding the cytotoxic patient effect? Any effect modification based on the clinical syndrome patients present with in the model? 

Bleeding grades in the multivariate analysis (303) were associated with lower odds; please present as normally done in a multivariate analysis assuming the odds is lower but not below 1..? 

Line 311: why were patients included who probably did not have a snakebite? 

Line 313: who decided the scars were unsightly? And who decided the amputations were minimal and the functional limitations? Patient reported?

Discussions: Line 322 – 352 are not related to the study findings and read as a general background. 

Line 397: female gender was associated,…. We did not have a clear explanation for this finding. Which leads to the earlier comment/question asking the authors to answer the primary research question and some secondary questions but reduce the post hoc analyses? 

How do authors know the study also confirms that IPA no more than other antivenoms is unable to stop or slow the progression of edema? There was no comparison to other antivenoms and the follow-up method seemed not adapted to assess impact on cytotoxic lesions as these focused on bleeding and neurotoxicity.

Author contributions: 

Can authors explain the author contributions a bit more? The study took place in Cameroon but the conceptualization and methods were developed by two team members in Paris only. Was there no codevelopment with the team in Cameroon? And the only software used is STATA and REDCaP but one coauthor has software as the only role? 

Baseline factors associated with neurotoxicity N-23 but there are 21 variables in the model and many of them require dummy variables which further hampers power. Which power did the authors expect with this combination? With such limited power, why do you add the different first line home treatments whereas that question does not fit one of your main research questions (outcomes after AV and tolerability of AV?) Same applies to the model with haematotoxic ‘troubles’?

Reviewer #2: Authors have conducted a thorough investigation of IPA in SSA. These findings are important to the region and the results are promising. Some minor comments are found which I believe can be addressed for clarity.

Reviewer #3: The article is interesting, however it is necessary to clarify how the coagulation correction was evidenced at two hours taking into account the absence of initial values and two hours later since it does not show laboratory data

PLOS authors have the option to publish the peer review history of their article (what does this mean?). If published, this will include your full peer review and any attached files.

Reviewer #1: No

Reviewer #2: Yes: Norman L. Beatty, MD, University of Florida College of Medicine, Gainesville, Florida, USA

Reviewer #3: No
---

## [Decision Letter · Decision Letter 1]

9 Oct 2023

Dear Dr. Chippaux,

We are pleased to inform you that your manuscript 'Real life condition evaluation of InoserpTM PAN-AFRICA antivenom effectiveness in Cameroon' has been provisionally accepted for publication in PLOS Neglected Tropical Diseases.

Best regards,

Wuelton M. Monteiro, Ph.D.

Section Editor

Manuela Pucca

Guest Editor

Reviewer's Responses to Questions

**Key Review Criteria Required for Acceptance?**

**Methods**

-Are the objectives of the study clearly articulated with a clear testable hypothesis stated?

-Is the study design appropriate to address the stated objectives?

-Is the population clearly described and appropriate for the hypothesis being tested?

-Is the sample size sufficient to ensure adequate power to address the hypothesis being tested?

-Were correct statistical analysis used to support conclusions?

-Are there concerns about ethical or regulatory requirements being met?

Reviewer #4: (No Response)

**Results**

-Does the analysis presented match the analysis plan?

-Are the results clearly and completely presented?

-Are the figures (Tables, Images) of sufficient quality for clarity?

Reviewer #4: (No Response)

**Conclusions**

-Are the conclusions supported by the data presented?

-Are the limitations of analysis clearly described?

-Do the authors discuss how these data can be helpful to advance our understanding of the topic under study?

-Is public health relevance addressed?

Reviewer #4: (No Response)

**Editorial and Data Presentation Modifications?**

Reviewer #4: Accept.

**Summary and General Comments**

Reviewer #4: I strongly recommend the publication of this highly pertinent study on snakebite treatments, which specifically examines the efficacy of an existing commercial antivenom in Cameroon, which is among the regions most afflicted by snakebites within the African population.

This study should be considered for publication in PLOS Neglected Tropical Diseases, as the authors have meticulously addressed all reviewer queries and have significantly enhanced the manuscript's quality, rendering it well-suited for publication.

PLOS authors have the option to publish the peer review history of their article (what does this mean?). If published, this will include your full peer review and any attached files.

Reviewer #4: **Yes: **Isadora Sousa de Oliveira

---

## [Editor Report · Acceptance letter]

30 Oct 2023

Dear Dr. Chippaux,

We are delighted to inform you that your manuscript, "Real life condition evaluation of Inoserp PAN-AFRICA antivenom effectiveness in Cameroon," has been formally accepted for publication in PLOS Neglected Tropical Diseases.

Best regards,

Shaden Kamhawi

co-Editor-in-Chief

Paul Brindley

co-Editor-in-Chief
